# Errors? Not Too Worrisome. Exploring the Effects of Errors in Speech Transcription

## ABSTRACT

This paper presents two user studies that investigate how errors that occur during speech recognition affect users' text entry performance and experience. For our work, we used a speech recognition system that injects believable errors in a controlled manner, and where users could fix errors by re-speaking a small part of their original utterance. Participants were asked to transcribe a set of phrases using our system, either with or without the insertion of errors, In the first study, we injected up to 33% errors, but saw no substantial results. Yet, participants commented consistently on the used phrase set, which did not correspond well with spoken English. Thus, we created a novel phrase set based on spoken phrases. In our second study, we used this phrase set and inserted errors into the speech recognition results with either 25% or 50% probability. The results showed that inserting errors in the speech recognition system had a significant effect on participants' perceived mental workload. In addition, we find that inserting errors increased the number of errors users made during the task. According to our findings, users have a fairly high tolerance for errors encountered in speech transcription.

**Index Terms:** Human-centered computing—Interaction design and evaluation methods—Speech recognition—Re-speaking;

## 1 INTRODUCTION

In today's digital era, text entry has become an indispensable aspect of both our social and professional lives. Hence, efficient text entry has become vital for effective communication. Commonly used state-of-the-art technologies to improve text entry efficiency include auto-correction and word prediction. However, these features are not always perfect. In fact, some studies have shown that auto-correction and predictive features rarely increase text entry speed significantly, due to the time required to manually fix wrong predictions, corrections, and/or the higher cognitive load required to fix such errors [5, 7, 17, 22]. When such errors occur, users also experience an increase in frustration as well as physical and mental workload [5, 22, 23].

Beyond predictive features and autocorrection for typing-based input, voice-to-text has become another widely used feature for text entry [10, 27]. With recent advances in Automatic Speech Recognition (ASR), voice assistant tools such as Google Home and Amazon Alexa have gained popularity, and today most digital devices support built-in dictation. A body of research has shown that entering text through voice can be much faster than typing [12, 16, 26, 30]. Yet, there has been a lack of studies that investigate how errors encountered during speech recognition can affect users' text entry performance and also their frustration.

The central objective of our work is to understand how users respond to errors encountered in speech transcription tasks. Accordingly, we present a quantitative analysis of user responses to different error frequencies, complemented by qualitative user feedback and observations. In addition, based on feedback from users when encountering traditional text entry phrase sets, we also created a new phrase set that is (better) suited for speech transcription tasks.

## 2 RELATED WORK

In this section, we review prior work that is relevant for our work. Previous research mainly focuses on hands-free techniques to correct errors, rather than how users respond to the errors.

### 2.1 Error correction with voice input

Error correction plays a critical role in text entry. Advances in error correction algorithms have enabled improvements in the efficiency of typing-based text entry and the users' experience [24, 28, 35]. Still, voice-based text input can afford much higher text entry rates compared to typing [8, 12, 16, 26, 30]. However, the level of acceptance of speech as a means of text input is not as high as typing, one of the major causes being the difficulty to correct text entry errors [18]. Previous work identified that using multi-modal input, i.e., multiple modes of input, might improve error correction in speech recognition [21, 30]. For example, combining voice input with a mouse or keyboard can aid error correction better compared to using voice-only [15]. In real-world text entry scenarios, there is a tendency to start with voice-based text entry first, and then switch to the keyboard to manually correct the text [14, 18]. This switch of modality can greatly impact the user experience, and is especially challenging for visually impaired users [8]. Azenkot et al. found that visually-impaired participants spent more than 80% of their time editing speech recognition results, with most edits using the backspace key and reentering text with the keyboard [8]. Similarly, sighted people spent 66% of their time editing the text on a desktop dictation system [16].

Unfortunately, accurate error correction via voice is more challenging than through typing, due to the linear and temporal nature of audio making it more prone to errors [11]. The major challenge in voice-based error correction is specifying the location and the size of the erroneous (part of the) phrase [9, 16, 21].

To address this issue, McNair and Weibel [20] proposed an effective error correction technique that only requires re-speaking the erroneous part of a sentence (or longer text). With this, users simply re-speak the erroneous word(s), e.g., correcting "I will bang it over tomorrow" by saying only "bring", instead of re-speaking the full sentence. Given that matching the re-spoken part to the original utterance can be challenging and can yield ambiguous results, this technique was further improved by Ghosh et al. [13] (more specifically their technique outlined in the appendix available in the ACM Digital Library). Their approach lets users provide a bit more context by re-speaking more than the erroneous part. The additional words enables the system to detect the incorrect part of the phrase more accurately, making error correction more reliable. In the above example, users could thus, e.g., say "bring it over" to correct the text in an unambiguous manner. We adopted this re-speaking technique of Ghosh et al. [13] for our evaluation. To ensure that the beginning/end of any speech act (be it the initial utterance or an error correction) was detected reliably, we combined it with minimal use of the mouse. We discuss the details of our implementation in the Apparatus section.

### 2.2 Measures of user performance and user experience

To assess participants' interaction with the system, we used two different types of measures: text entry efficiency and subjective experience measures. To quantify the entry efficiency, we measure

the entry speed, verification time, and accuracy of the entered phrase relative to the prompt. To assess the subjective experience, we measure participants' self-reported mental and physical demand, and their frustration levels. According to the results of previous studies on error correction, we expect to see impaired entry efficiency and an increase in perceived workload and frustration as the number of errors increases [5–7].

### 2.2.1 Entry Efficiency

The two most commonly used measures for text-entry efficiency are word per minute (WPM) and error rate (ER). WPM measures the speed of the text entry, with faster WPM indicating more entry efficiency. In voice-based typing, it also refers to how fast users speak. ER measures the accuracy of the text entry, where we compare the given text (in transcription tasks) with the final text users entered, with lower ER indicating more entry efficiency. Both WPM and ER have been frequently used in literature to measure text-entry efficiency [4, 5, 7, 8, 16]. We also added another metric called verification time (VF) to measure how long it took participants to decide that the result is correct, after they finished entering the text [6].

### 2.2.2 Subjective Experience

The NASA TLX is a well-validated questionnaire that was originally designed to measure workload in the military [23], but it has been applied in a variety of settings in HCI research [19]. The NASA TLX combines six different scales including mental demand, physical demand, temporal demand, effort, performance, and frustration. In our studies, we were only interested in the perceived mental and physical demand, and the frustration experienced by users, and thus used only the corresponding questions.

## 3 STUDY 1

Our first study used a between-subject design with the injected error rate (two levels, 0%, and 33%) as the independent factor. The dependent factors included entry speed (WPM), error rate (ER), as well as self-reported frustration, physical demand, and mental demand (NASA TLX). Participants were randomly assigned to one of the 0% error and 33% error conditions. We collected 29 phrases for each participant, for a total of 348 phrases.

### 3.1 Participants

Twelve participants (six females, six males), aged between 21-29 years old, with an average of 24.5 (SD = 2.15) participated in this study. All participants were either completing or had completed a bachelor's degree in an English-speaking university in Canada. All data were collected over Zoom. Participants generally shared their screen, except for two who did not agree to share the screen.

### 3.2 Apparatus

The experiment used a web application housed on a local university server. We implemented the system using HTML, CSS, JavaScript, and the Google speech recognition API [3]. Fig. 1 shows the interface of our system. Normally, such a system would show the most likely recognition result for the user's utterance, but we injected errors (with a controlled frequency) by showing the second-most likely result returned by the Google API, which yields a very believable misrecognition. For instance, instead of showing "How was your trip to Florida?" (the most likely result), the system would then display "How was your train to Florida?" (the second-most likely result). In the two experimental conditions, we injected such an error either 0% or 33% of the time. We chose 33% in this study as a compromise between being able to observe sufficient errors and avoiding excessive frustration. For error-correction through the re-speaking feature, we permitted users two attempts for transcribing each phrase. For the first attempt, users were required to speak the whole phrase in full, while the second attempt was reserved for correcting any errors that remained after the first attempt. Errors were only injected for the first attempt, and the system always showed the most likely result in the second attempt.

### 3.3 Task and Procedure

All phrases were randomly selected from the Enron MobileEmail phrase set [34]. We removed all punctuation marks in the recognition results, as they might introduce a confound in the dependent variables, which might undermine internal validity [7, 29].

At the start of this study, participants were allowed to choose the most appropriate accent among English-US, English-UK, English-India, and English-Canada. Participants were then asked to speak each phrase that appeared on the screen. Participants clicked on "Start Recording" to start recording their utterance (Fig. 1 left), followed by a click on "Stop Recording". If the speech was transcribed incorrectly, then they could repeat part of the sentence by clicking on the "Start re-recording" button on the same page (Fig. 1 right), and then again "Stop Recording".

**Instruction on the Re-speaking Feature** We asked participants to (generally) re-speak the incorrect phrase from at least one word before the incorrect word and end at least one word after the incorrect word. If the correction involved the ending or starting word, then they were asked to repeat two words after or before. The system did not enforce this instruction, instead the algorithm matched whatever participants used to correct an error. Participants were only given a single error correction attempt for each phrase, after which they had to proceed to the next phrase.

After participants completed all 29 phrases, they completed a NASA TLX 7-point Likert scale questionnaire (with 1 = lowest, 4 = neutral, and 7 = highest). Each participant was asked to self-report their subjective experience, followed by a brief interview at the end to assess participants' familiarity with speech-to-text systems and their experience with the re-speaking interface.

### 3.4 Quantitative Results

To analyze the error rate, we compared participants' final submitted transcription results with the phrases they were given. We considered three types of errors in our study: user-caused errors, the errors we injected, and the errors that resulted from the occasional glitch in the speech-recognition system (i.e., the Google Speech-to-text API). For our study, we are specifically interested in the first type—the errors that are caused by the insertion of the second type.

Fig. 2 shows the results for all dependent variables. Overall, error rates for participants in the 33% error condition (M = 29.9%, SD = 13.7%) were higher than with 0% errors (M = 23.6%, SD = 6.3%). However, there was no significant difference between the two conditions, t(7.03) = 1.02, p = 0.34.

Overall, participants exhibited a higher WPM in the 0% error condition (M = 119.90, SD = 13.77) compared to 33% errors (M = 116.08, SD = 10.10). Yet, a Wilcoxon test revealed no significant difference for entry speed (Z = -1.04, p = .30).

In the 0% error condition (M = 1.50, SD = .55) participants reported lower physical demand than with 33% errors (M = 1.67, SD = 1.63), but the difference was not significantly different (Z = -.76, p = .44).

The mental demand for participants in the 33% error condition (M = 3.50, SD = 1.60) was higher than with 0% (M = 2.50, SD = 1.52). However, the result was not significantly different, t(0.74) = 1.02, p = 0.48.

Overall, frustration for participants in 33% error condition (M = 3.17, SD = 1.72) was higher than in the 0% condition (M = 1.33, SD = .52). A Wilcoxon signed-rank test between the 0% and 33% error conditions revealed only a marginally significant difference for frustration (Z = 1.95, p = .05, $\eta_p^2$ = .03).

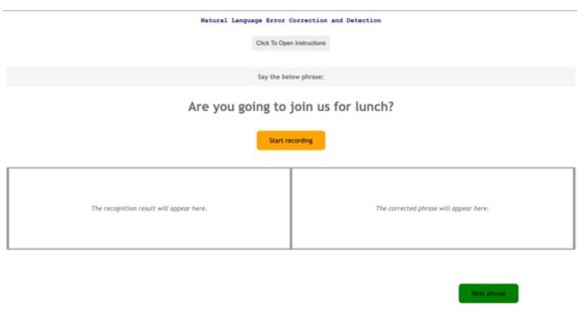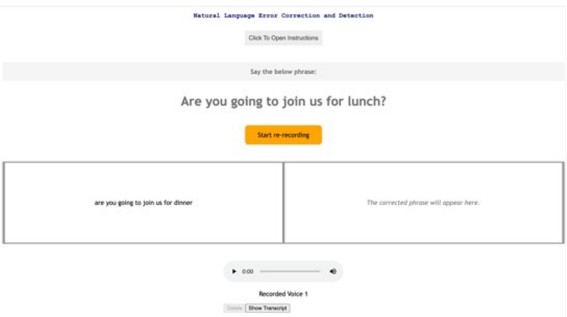

Figure 1: Interface of the speech-to-text system in the first study.

### 3.5 Qualitative Results

All participants reported at least some level of familiarity with speech-to-text tools during our interview. When we asked about their experience with our system compared to other systems they had used before, participants had very diverse responses, regardless of which condition they experienced. For example, a participant in the control condition reported the system as *"very dumb, needs more development"*, whereas another participant in the experimental condition said: *"It has better accuracy, understood me better."* When we asked them about the most challenging part of the study, a majority–7 out of 12 participants–reported disliking the phrases we used in the study. Some said the phrases felt weird and unnatural, took them a long time to read, and they also made more mistakes. Others mentioned that some of the phrases were too long.

### 4 DISCUSSION

One of the most striking findings from our study was the lack of significant difference, even though we induced a non-trivial amount (33%) of errors. While we found some marginal significance for frustration, the effect size was small, which indicates that the difference between the two groups was not substantial. Yet, one of the key takeaways from the qualitative result was the apparent inappropriateness of the phrase set for a task that involves speaking. Based on our participants' responses, we believe that the properties of the phrase set could be one of the most likely explanations for our results.

#### 4.0.1 Spoken vs. written language

Most current text entry studies employ transcription to evaluate text entry efficiency. Yet, the most widely used phrase sets in such studies were all designed for typing, not for speaking [19, 31, 34]. Relative to text transcription, text composition is more representative of real-world text entry scenarios. Although the composition task has higher external validity than the transcription task, the latter can outperform the former in experimental evaluations due to its higher internal validity and lower variability [33]. A recent study [12] let participants compose their own phrases, and use them for transcription tasks with other participants, which increased the logistic effort substantially. According to Foley et al., [12], phrases for transcription tasks have to fulfill the following characteristics: memorable (users can enter a phrase after the prompt without referring to it), representative (resemblance of the actual text entered by people), and replicable (the phrase set is publicly available).

Based on the results of the first study reported above, it seems that a set of written phrases might not be suitable for speech-based text entry evaluation. Thus, we believe that using such an un-representative phrase set might introduce confounds into a study. Spoken and written language contrasts in various aspects: spoken language is less abstract, has more finite verbs, and has fewer nouns of abstraction. There is also a contrast in syntax and sentence structure, and in terms of the manner and speed of production [4]. Moreover, entering text by tying versus speaking can lead to very different experiences for users [11, 26]. Neuroscience research also found that written and spoken language involves two distinct systems that are controlled by different parts of the brain [23]. Therefore, phrase sets for spoken and written language are likely not interchangeable.

#### 4.0.2 Phrase set for speech-based text entry

We collected a new phrase set from spoken English corpora. Unlike in the majority of text-entry research, where the phrases were (largely) selected from written text, such as the Enron MobileEmail phrase set [25], we selected our phrases only from spoken English corpora, using the Santa Barbara Corpus of Spoken American English (SBCSAE) [2] and https://www.english-corpora.org [1]. We initially selected about 1000 phrases, but removed all phrases that contain inappropriate language, uncommon names, or complex vocabulary, to avoid potential confounds within speech-based text entry evaluations. We ended up with 784 phrases, all with generally short to medium length, on average 11.5 words (SD 4.14, ranging from 6 to 27).

### 5 STUDY 2: RE-EVALUATION OF THE SYSTEM USING THE NEW PHRASE SET

After finding a generally negative opinion of the phrases used for study 1, we decided to re-conduct our study with the new phrase set we collected.

### 5.1 Study Design

In this study we used a within-subject study design, with every participant participating in all three conditions, which only varied in terms of the frequency of injected errors. The order of the conditions was counterbalanced among participants. In this second study, we injected an error 0% (control condition), 25%, or 50% of the time in the three conditions.

### 5.2 Apparatus

We adapted the system from Study 1, with an improved user interface (see Fig. 3), and also added support for all sixteen English accents supported by Google Speech-to-Text API. We also included the option for participants to either use the mouse/trackpad to control the buttons or press keys on the keyboard (space bar for start/stop recording and right-arrow key for the next phrase).

### 5.3 Participants

We recruited eighteen participants from the local university (mean age = 23 years, SE 5.12, 15 female) via on-campus flyers and word-of-mouth. All participants reported intermediate to professional levels of English-speaking skills. Most participants had had some experience with speech-to-text systems, and only one reported that

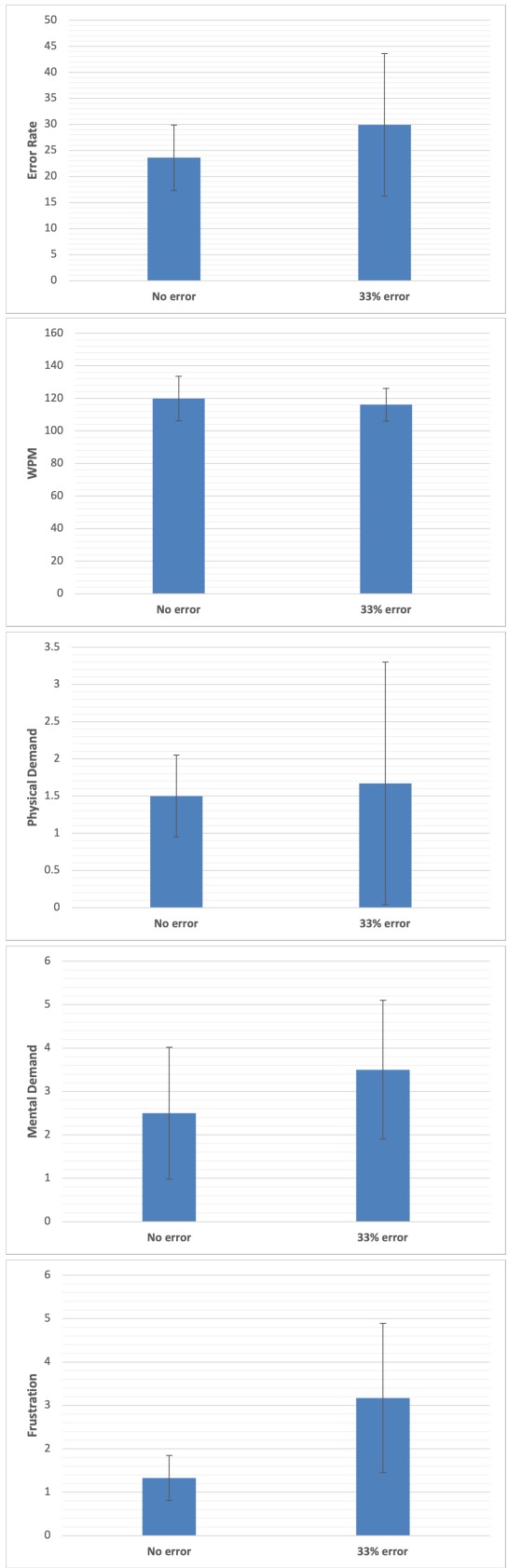

Figure 2: Average responses for Study 1. From top to bottom: ER, WPM, Physical Demand, Mental Demand, Frustration.

they had never used such systems before. Ten participants reported English was not their first language.

### 5.4 Procedure

After the consent procedure, we gave participants detailed verbal instructions for the task, followed by a brief demonstration of how the system works. Participants then were given practice trials until they demonstrated an adequate understanding of the system, which all participants were able to achieve within five phrases. Due to the limitation of the Google Speech-to-Text API, participants were told to ignore all punctuation marks, contraction issues, and spelling inconsistencies such as Canadian versus US spelling. Each participant performed a block of phrases in each of the three conditions, with each block containing 30 phrases. Thus, each participant completed 90 phrases. All participants experienced the phrases in random order, with the order of conditions counterbalanced across participants. In total, we collected 90 x 18 = 1620 phrases.

At the start of the experiment, participants were asked to choose their preferred English accent among sixteen English-speaking countries and regions, as supported by the Google Speech-to-Text API. As in study 1, participants then were asked to speak each phrase that appeared on the screen. Participants clicked on "Start Recording" to start recording their utterance, followed by a click on "Stop Recording". If the speech was transcribed incorrectly, then they could repeat part of the sentence by clicking on the "Start re-recording" button on the same page (Fig. 3), and then again "Stop Recording".

We asked participants to (generally) re-speak the incorrect phrase from at least one word before the incorrect word and end at least one word after the incorrect word. If the correction involved the ending or starting word, then they were asked to repeat two words after or before. The system did not enforce this suggestion, instead the algorithm matched whatever participants corrected. Participants were only given a single error correction attempt for each phrase, after which they had to proceed to the next phrase.

Immediately after each block, we asked participants to fill out a NASA TLX questionnaire assessing their perceived workload on the respective block. After they completed all three blocks, we conducted a short, semi-structured interview with participants regarding their overall experience with the system.

### 6 RESULTS

We summarize the results of our second user study in this section. We removed all log entries where the system had not worked correctly due to a small implementation issue (less than .03% of the data). We then ran a one-way repeated-measure ANOVA with Tukey's Honest Significant Difference (HSD) for all quantitative analyses. We use partial eta squared to calculate the effect size. We begin by examining participants' overall performance in the three conditions in terms of entry speed, verification time, and error rates. We then examine the participants' subjective experience of the system using NASA TLX. Finally, we review the qualitative observation and interview results.

### 6.1 Entry Efficiency

We used the word-per-minute metric (WPM) to measure participants' entry speed. Speed was calculated based on the number of words participants entered and their total completion time. We did not find significant differences across conditions, $F(2,34.02) = 1.20$, $p = .31$. See Fig. 4a.

For the analysis of the user-caused error rate, we disregarded any spelling variations due to different English spellings, such as "honour" versus "honor"; contractions such as "we're" versus "we are"; and common word combination issues such as "on to" versus "onto". The results indicated a significant difference across three conditions, $F(2,111.9) = 15.43$, $p < .0001$, $\eta_p^2 = .23$. Specifically,

Figure 3: Interface of the speech-to-text system used in the second study.

the posthoc test revealed significant differences between the no-error and 25% conditions (p < .0001) and the no-error and 50% conditions (p < .001), but not between the two error conditions (ps > .05). With a mean error rate of .17 (SE .09), the error rate in the no-error condition, was lower than in the 25% condition (.24, SE .06) and in the 50% condition (.29, SE .11). See Fig. 4b.

We also logged the verification time (VT), which refers to the time that participants took to review the phrases. We logged three VTs, with VT1 indicating the time elapsed from when participants saw their first recognition result and when they clicked either the "Start Re-recording" or the "Next Phrase" button; VT2 indicating the time elapsed from when participants saw the phrase resulting from the second speech recognition, i.e., the error correction, and when they clicked on the "Next Phrase" button; and the total VT adding these two times together. The results indicate that there were no significant differences across all conditions for VT1 (Fig. 4b) and the total VT (Fig. 4e, ps > .05). Fig. 4d illustrates the statistically significant difference for VT2, F(2,34) = 4.44, p = .02, $\eta_p^2$ = .009. Post-hoc test revealed the difference was only significant between the no-error and 50% conditions (p = .02), but not between the other pairs (ps > .05).

## 6.2 Subjective Experience

To assess the participants' subjective experience, we asked participants to rate their perceived workload (NASA TLX) for the three conditions on a 7-point scale (with 1 = lowest, 4 = neutral, and 7 = highest). Fig. 5a shows the significant difference in participants' self-reported mental demand across conditions, F(2,34) = 3.69, p = .04, $\eta_p^2$ = .03. Yet, there was no significant differences for physical demand (Fig. 5b) nor frustration (Fig. 5c), all ps > .05. A posthoc test revealed that only the difference between the no-error and 50% conditions to be significant(p = .03), but not between the other conditions (ps > .05). The average mental demand rating for the no-error condition (mean 2.22, SE 1.06) was also significantly lower than the 50% condition (mean 2.72, SE 1.34).

## 6.3 Qualitative Observations

Although the Google Speech-to-Text API supports sixteen different English accents, 15 out of 18 participants selected American or Canadian English, and only three participants chose to actively look for the variant that corresponds to their native language. In the end, only a single participant selected a non-North-American accent (Indian accent). The other two participants stated that they did not find their preferred accent listed as an option. Some participants commented that they did not know whether they had an accent or not.

During the task, we observed that participants tended to review the given phrase before they started speaking it, but this tendency only occurred when they encountered longer phrases. Also, it was not uncommon for participants to overlook errors in their recognition results, especially for shorter words such as "and" or "the", especially when transcribing longer phrases. For such small words, they occasionally failed to notice errors related to them and then proceeded to the next phrase, not even using the second step, i.e., the correction attempt.

Although we provided the option to control the buttons using either keyboard or mouse/track-pad, we observed a strong preference for the mouse.

## 6.4 Qualitative Interview

Overall, participants rated their experience with the system between neutral to positive. More than half of the participants directly mentioned that they liked the re-speaking feature, as it was convenient and time-saving. Since 17 out of 18 participants reported occasional usage of speech-to-text systems, we focused on the comparative experience of our system with other systems they had used in the past. When we asked them to name a few systems they have used, iOS built-in dictation, Google's pages/apps (Translation, Search, and Google Home), and Microsoft Word were the most frequent answers. There was a tendency to use such systems only when their hands were busy, such as while driving or cooking. Since the majority of the participants were university students, using such systems for lecture transcription or essay writing was also a common answer. For social purposes such as texting, there was a clear preference for typing, as they perceived it to be more accurate. In terms of accuracy, 11 out of 18 participants commented positively on the accuracy of our system, and 4 of them even thought the system was more accurate than the speech-to-text systems they had used in the past.

On the other hand, we received diverse responses in terms of negative feedback. For example, participant 2 said *"It was hard that I had to only use voice to correct the sentences, I wish I could type."* Participant 17 said *"When I have a longer pause, the system won't pick up properly, or add extra words to my sentences. When there was slang in the sentence, it keeps changing to formal English. For example, I say 'gonna' but the system keeps recognizing it as 'going'. I prefer a more slang-friendly system."* Two participants also mentioned that the re-speaking feature did not work well when there were two incorrect parts in the first recognition result (which was indeed a limitation of our implementation, as we supported only a single correction "action").

We also asked participants what they thought was the most challenging part of the task, and two participants responded that it was challenging for them to get used to the re-speaking feature. Other participants had more diverse responses, for example, participant 3 said: *"When sentences were long, I had to read it before I started recording. If I didn't go through it first, I might not pronounce some of the words very well, especially since English isn't my first language."* Participant 9 said, *"I had to make sure I speak really slowly*

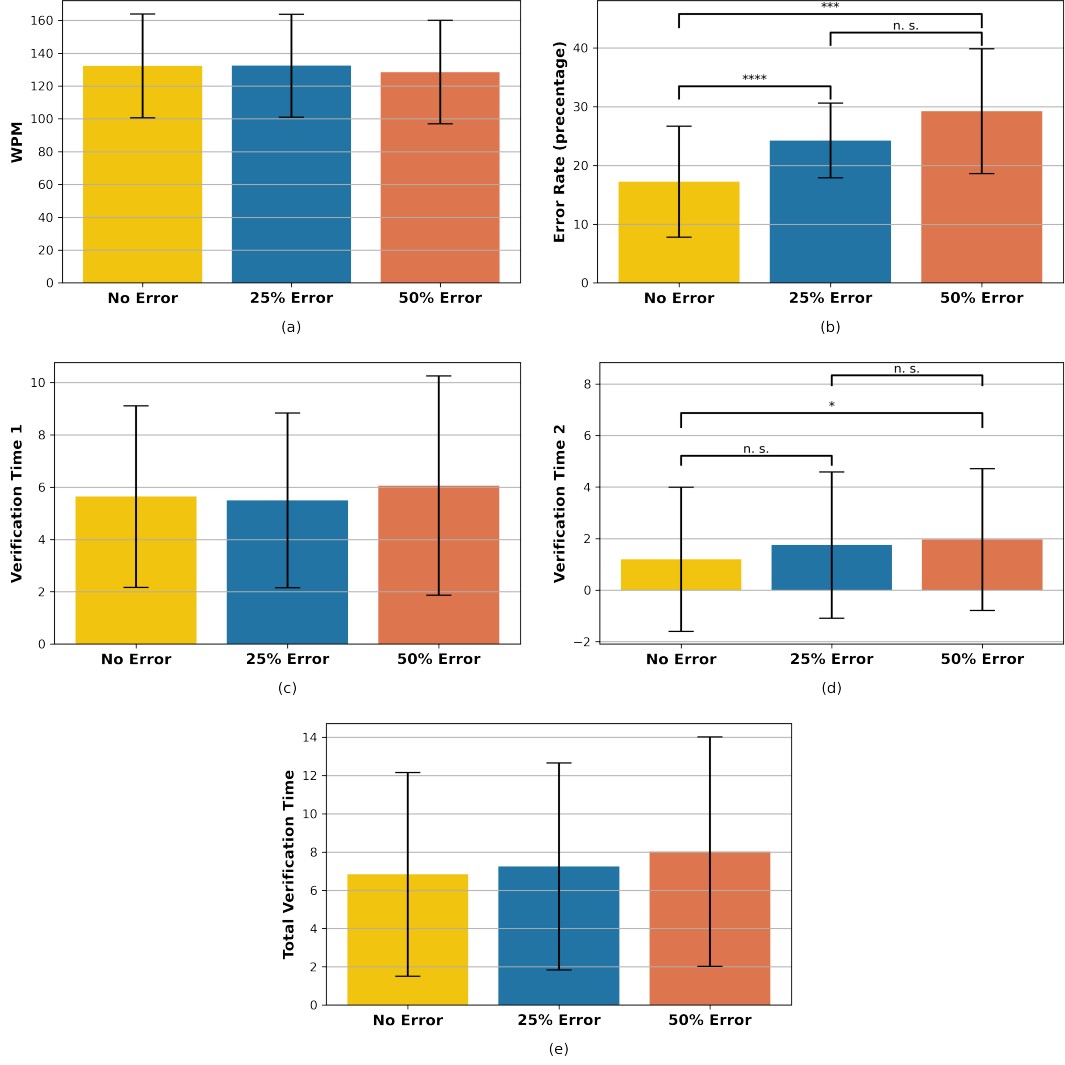

Figure 4: a) Average words per minute (WPM), b) average error rate (ER), c) first verification time in seconds (VT1), d) second verification time in seconds (VT2), and e) total verification time in seconds (VT).

*and clearly for the system to work. This was the first time I use a speech-to-text system, and I don't see the need to use it in the future."* Some participants also mentioned that sometimes the re-speaking failed to correct the phrase as they intended. Beyond the limitation to a single correction action mentioned above, such (rarely occuring) system failures sometimes duplicated text rather than replacing it; added extra words to the phrase; or failed to correct the phrase when the erroneous part was at the very beginning of the phrase. Such systems failures occurred in less than 1% of all trials.

Participant 18 said *"When there was an error, I first had to pinpoint the error and then changed my speech style for it to recognize. But this is just how I talk, I don't know what to change. I feel like technology should adapt to us, instead of the other way around."* Some participants also mentioned the difficulty of getting homonyms (words with the same pronunciation but different meanings) to be recognized correctly, such as "sight" versus "site".

Additionally, we asked participants if there are any other ways they could think of to edit the phrases using voice. Two participants said they would rather just re-speak the entire phrase, instead of having to locate the erroneous part and correct it. Four participants

directly mentioned using voice instruction to correct the phrase, and participant 5 suggested: *"It might be better to have something like the 'Hey Siri feature', you can then just say something like 'Correct this word to that.'"* Two participants stated that they would like to have a third attempt to correct the phrases, *"Especially when there are two incorrect parts"* (participant 15). Participant 11 suggested that: *"Maybe you can say a keyword to put you in the error correction mode. For example, you say 'correction [keyword]: A [wrong word] to B [correct word]'."*

We did not receive any negative feedback regarding the phrase set. Some participants commented that the given phrases felt often more conversational than the transcribed result (also because the speech recognition changed their recognition result to more formal English).

## 7 DISCUSSION AND FUTURE DIRECTIONS

Among all conditions and all text entry efficiency metrics we measured, we only observed a significant difference with a large effect size on the user-generated error rate, with no-error condition producing the lowest amount of errors, relative to the two other conditions.

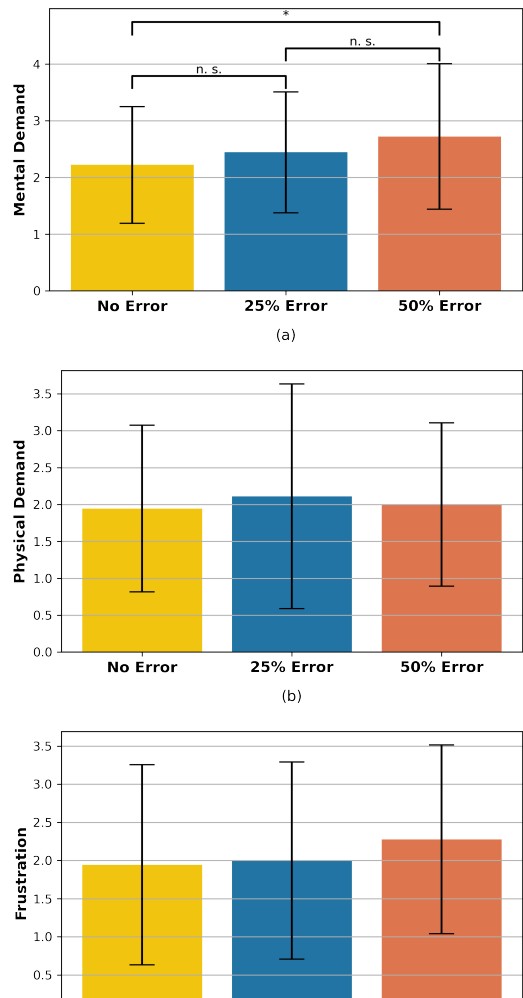

Figure 5: NASA TLX scores for a) Average mental demand, b) average physical demand, c) average frustration.

It seems obvious that the final text tended to contain more mistakes when more errors were inserted by the system. Since we only inserted errors during the first attempt, we can conclude that, as system-generated errors increase, a re-speaking feature is not sufficient enough for error recovery.

Still, a larger percentage of injected errors in the system did not lead to longer verification times or a slower entry speed. Although we discovered a significant difference in the second verification time (VT2) between the no-error and 50% error conditions, the effect size was small, which might indicate a potentially spurious result. The increased second verification time might also indicate that users were more careful during the text correction phase when a larger number of errors were present.

Given these results, the most surprising outcome of our study is that even though we injected a relatively large amount of errors 50%, users did in general not perform substantially different with the system. In other words, they encountered on average an error on every second phrase.

In regards to the NASA TLX results, we only observed a significant difference in mental demand between the no-error and 50%

conditions. Still, the effect size was rather small for this result, too. In fact, participants' ratings tended to be generally low for the NASA TLX scores (Fig. 5).

Moreover, we found a positive correlation between the length of the given phrase and WPM, $r(425.23) = .46$, $p < .002$, suggesting a moderate correlation between the two variables. This might suggest some link between the phrase length and WPM. Yet, we expect to see this relationship plateau at some limit.

All participants in our study had never used a re-speaking feature before. Although we did receive some positive feedback, some participants also mentioned that they would prefer re-speaking the entire phrase, rather than checking the recognition and locating the erroneous part.

From our qualitative results, we noticed that participants generally exhibited a relaxed attitude toward the errors they encountered. When we asked for a comparison between our system and other voice-to-text systems they had used, participants often brought up scenarios where other systems make errors. The general consensus among participants was along the lines of *"Although your system sometimes makes errors, other systems also make errors—even more errors. So your system isn't that bad."* This suggests that participants are more used to errors in speech transcription systems.

From the interview results, we also found that users noticed some imperfections in the re-speaking feature, especially surrounding issues with the matching algorithm (even though this occurred rarely). Indeed, a re-speaking feature can be prone to alignment issues [32], which can be confusing for users as they then have no indication of how much text they need to re-speak [11].

Overall, our results suggest that the insertion of errors in speech transcription might have some effect on the entry efficiency and perceived workload. However, in contrast to our expectations, the effect seems to be small. The results suggest that, compared to typing-based text entry tasks, users seem to have a higher tolerance to errors in speech transcription tasks.

In the future, we plan to re-evaluate our system using other error correction methods, such as direct voice commands. We also plan to include the third attempt for the error correction, as suggested by some participants in the interviews.

## 8 CONCLUSION

Through two user studies on speech-based transcription, we explored the effect of errors on entry efficiency and subjective experience, using both quantitative and qualitative measures. We evaluated the efficiency of the re-speaking feature for error correction. Our results identify that users are more likely to make errors in their final submitted recognition results when more errors occur in the system. We also found that users might be more likely to have higher frustration and mental workload as the rate of errors increased. Based on our findings for the second verification time (VT2), we also identified that the re-speaking feature is likely not sufficient for error recovery, especially when a high number of errors occurs in the system.

Overall, our findings suggest users have a fairly high tolerance for errors in speech transcription. Our findings are thus potentially good news for speech interface designers and programmers, as a limited amount of errors in the systems might not affect the user experience in a substantial way.

### ACKNOWLEDGMENTS

We would to thank Afshan Ahmed for the initial prototype of the system, and Mohammad Rajabi Seraji for providing help with fixing bugs in the system.

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
