# OpenReview forum: "Errors? Not Too Worrisome. Exploring the Effects of Errors in Speech Transcription."
_graphicsinterface.org/Graphics_Interface/2023/Conference_SD — Submitted to GI 2023 - second deadline_

### Official Review · Reviewer_gf7c · 2023-04-04
**Good perspective of looking at the user experience of ASR, but needs a bit more work**

**Rating:** 4
**Confidence:** 4

**Review:**

This article discusses two experiments that examine the impact of speech recognition errors on users' text entry performance and experience. The authors used a speech recognition system that introduces realistic errors in a controlled manner, and users were given the option to correct errors by re-speaking a portion of their original statement. Participants were asked to transcribe a set of phrases using the system, with or without the introduction of errors.

The first study tested errors of up to 33% but did not yield significant results. However, users consistently commented on the phrase set used, which did not match spoken English well. Therefore, the authors created a new phrase set based on spoken phrases for the second study. In the second experiment, errors were introduced into the speech recognition results with either 25% or 50% probability using the new phrase set. The findings revealed that introducing errors in the speech recognition system significantly impacted users' perceived mental workload and increased the number of errors made during the task. Despite this, the study found that users had a high tolerance for errors in speech transcription.

One major issue I found from this paper was about how authors used NASA-TLX. As authored mentioned, the NASA TLX is a well-validated tool to measure the human subjective mental workload of a task. However, the subscales of six dimensions cannot be analyzed independently from other dimensions in NASA TLX as the tradeoff between dimensions must be considered to be analyzed for the total subjective mental workload. With this regard, it is suggested to include and report all other subscale values and the weighted scores to compare as well as to be tested for using NASA TLX to have the claimed validity. In addition, since the WPM was measured, why was Temporal Demand not part of the concerned dimension?

If the original intention is to use the Raw TLX, without pairwise comparisons, the evaluation can still be done under the claim of the named method instead. This should be noted as the dimensions of NASA TLX correlate with each other. However, it would be similar to a regular Likert-type scale questionnaire to survey user experience and measure frustration. Therefore, I would suggest reframing the work around this perspective instead of NASA TLX.

More comments

- I wonder if the rate of accuracy of Google Speech API was considered in the study, before the injection of errors. I can see the error was categorized as a ‘glitch’ but the rate of this can play one role- How was this measured? or was it a part of the considerations in comparing error rate?
- Study 1 had two injected error rates (0% and 33%). Why was 33% chosen to be level for the treatment condition? It would be great to have any supporting references. Similarly, unsure how 25% and 50% were chosen for study 2.
- How was the qualitative data analyzed? Was it using open coding? thematic analysis? If this is a narrative analysis, what were the categories? How many comments were in the categories? what was the importance of the comments/coda?

---

### Official Review · Reviewer_Pmon · 2023-04-20
**Studying the impact of error rates on user performance with speech recognition**

**Rating:** 4
**Confidence:** 3

**Review:**

This submission presents findings from two lab studies examining how injected error impacts user performance with a speech recognition system.

Given the prevalence of speech interfaces, understanding how to help users cope with errors is an important problem.  A challenge with this submission is that it is difficult to discern the specific research questions it aims to answer and why answering these questions would provide an important complement to the existing literature.

Two studies are reported on a speech recognition system that has a particular approach to error correction:  allowing users to re-speak fragments of the phrase by providing 1-2 words of context before and after the erroneous text.   Rather than comparing this approach to another alternatives, the studies compare the same approach with differing levels of injected error rates.  The findings are unfortunately limited in their scientific value in that we mainly learn that it is hard for users to work with the system when more errors are injected.  In my view, this alone isn’t sufficient to warrant a standalone publication.

Of the two studies, the first seemed like a good pilot study, but with 12 participants in a between-subjects design (6 per condition) it is difficult to interpret the results.  It is possible, for example, that individual variability dominated any condition effect.

A second listed contribution is in developing a phrase set that is more suitable to speech recognition studies than a state-of-the-art set. This is an interesting and potentially highly useful contribution; however, this component of the work isn’t sufficiently developed that other researchers could confidently use and build on the findings.  A more in-depth study of phrase set suitability for speech recognition research would likely be very well received.

There are several aspects of the reporting and presentation that would benefit from additional clarity and justification in future submissions:

-	Why were errors only injected in the first attempt?  How does this impact the external validity of the findings?

-	How were error rates chosen?  For example, are these error rates representative of the state-of-the-art?

-	I wasn’t clear on why the punctuation would introduce a confound.

-	With the “Choose Your English Accent” did participants see a list of options or was this a free-form textbox (Fig3)

-	What do the error bars in the graphs represent?

-	I would recommend reducing the space devoted to the qualitative findings.  This type of focused, lab-based text-entry study doesn’t typically lend itself to rich qualitative insights.

-	What is considered a large vs. small effect size?  I might have misunderstood the text, but there seem to be some inconsistencies in the discussion as to whether the findings have large or small effect sizes.  Effect sizes should be provided for the non-significant findings as well.

-	Acknowledgements should be removed for anonymous peer review.

---

### Official Review · Reviewer_e13i · 2023-04-21
**focus on study 2?**

**Rating:** 6
**Confidence:** 3

**Review:**

The goal of this paper is to investigate the impact of transcription errors on users' text entry performance and experience. The authors present two studies that utilize a speech recognition system to introduce believable errors in a controlled manner, and participants are asked to transcribe a set of phrases with or without error insertion. The results indicate that as error rates increase, users are more likely to make mistakes in their final submitted recognition results, and may experience higher levels of frustration and mental workload. Additionally, the researchers assessed the effectiveness of a re-speaking feature for error correction but found it inadequate when multiple errors occurred.

Overall, the paper is well-written and engaging, particularly the detailed discussion of Study 1. However, it may be more appropriate to place greater emphasis on Study 2 to avoid confusion about the main findings. If the participants overwhelmingly dislike the phrase set because it’s so challenging, would that reduce the efficacy of such study? Is it appropriate to spend several pages describing the study? Would it make more sense to focus on the Study 2?

Although the study's limitations are acknowledged, the authors' focus on examining only the effects of errors on text entry in a single task appears oversimplified, given that users rarely type for the sake of typing alone. In more complex interactions such as chats, emails, or essays, text entry is only one aspect of the experience. As such, it would be interesting to explore how incorrect transcription may affect other aspects of the user experience, such as the ability to add emojis or stickers and the effects of accidentally sending the wrong message.

In summary, this paper provides valuable insights into the effects of transcription errors on users' text entry performance and experience. However, the authors may wish to focus on the findings of Study 2 and consider the broader implications of their research in the context of more complex user interactions.